# Assessing Fermentation Quality, Aerobic Stability, In Vitro Digestibility, and Rumen Degradation Characteristics of Silages Mixed with Sweet Sorghum and Aerial Parts of Licorice

Feng Chen [1,2], Jiao Wang [1], Sujiang Zhang [1,*], Abdul Shakoor Chaudhry [3] and Hassan Khanaki [4]

[1] Key Laboratory of Tarim Animal Husbandry Science and Technology, College of Animal Science and Technology, Tarim University, Alar 843300, China; chenfengdky@163.com (F.C.); wang20220910@126.com (J.W.)
[2] College of Animal Science and Technology, Northwest Agriculture and Forestry University, Yangling 712100, China
[3] School of Natural and Environmental Sciences, Newcastle University, Newcastle upon Tyne NE1 7RU, UK; abdul.chaudhry@newcastle.ac.uk
[4] Faculty of Veterinary and Agricultural Sciences, The University of Melbourne, Dookie College, VIC 3647, Australia; h.khanaki@unimelb.edu.au
* Correspondence: zsjdky@126.com; Tel.: +86-152-9230-0910

**Abstract:** (1) Aim: This study aimed to evaluate the fermentation quality, chemical composition, aerobic stability, in vitro digestibility, and rumen degradation characteristics of silage mixtures with different ratios of sweet sorghum (SS) and aerial parts of licorice (LC). (2) Methods: Five mixtures were produced on a dry matter (DM) basis: (i) 0%SS + 100%LC (0%SS); (ii) 25%SS + 75%LC (25%SS); (iii) 50%SS + 50%LC (50%SS); (iv) 75%SS + 25%LC (75%SS); and (v) 100%SS + 0%LC (100%SS). First, the chemical composition of the silages was measured before and after fermentation. Next, the aerobic stability, dynamic microbial colonization and dynamic volatile fatty acids of the mixed silage after fermentation were determined for 0, 5, 10, 15, 20, and 25 days. Finally, the parameters related to gas production and the characteristics of the gas production were determined. At the same time, the rate of degradation of the chemical composition of the mixed silage in the rumen was studied. (3) Results: (a) As the proportion of SS increased, pH, ammonia, butyric acid, acetate, and aerobic stability showed a decreasing trend, but lactic acid content gradually increased. (b) The content of the fermentation and gas production parameters were significantly higher in 100%SS and 50%SS than others ($p < 0.05$). (c) The rate of degradation of DE, ME, Neg, DM, CP, ADF, NDF, and ADL of 50%SS in the rumen of sheep was significantly higher than others ($p < 0.05$). (4) Conclusions: In conclusion, ensiling SS and LC mixtures can improve silage quality, especially if the SS and LC are ensiled together at a ratio of 50:50.

**Keywords:** sweet sorghum; licorice; mixed silage; fermentation; aerobic stability; rumen; degradation characteristics

## 1. Introduction

The shortage of high-quality roughage resources is the main factor restricting the development of the sheep industry in many developing countries. Southern Xinjiang is the largest saline soil area and desertification area in China, which has relatively little natural rainfall and poor soil. Due to natural limitations, high-quality roughage resources in southern Xinjiang are extremely scarce. There is $0.067 \times 10^8$ hm$^2$ of marginal land in Xinjiang, among which the total area of saline land is $1.336 \times 10^7$ hm$^2$, accounting for 36.8% of saline–alkali land area in China [1]. The saline soil ecosystem is very fragile and the secondary soil salinization is serious in Xinjiang. The main solution is to popularize the cultivation of saline-tolerant forage crops.

Sweet sorghum (SS) is a promising forage that grows in arid, semi-arid, and high-salinity areas [2]. It is also known to be stress-resistant, drought-tolerant, and highly water-use efficient [3]. These excellent characteristics are very suitable for promoting planting in the southern Xinjiang region. SS is a resilient $C_4$, which has the characteristics of high fermentability, nutrient digestibility, and palatability [4,5]. However, the crude protein content in SS silage is insufficient to maximize growth efficiency in most ruminant production systems [6–8].

Licorice (LC) is a saline-resistant, drought-tolerant perennial herb of the legume family [9]. Licorice root and its extract are important Chinese herbal medicines with high medicinal value. It is mainly found in China, Mongolia, Central Asia, and Russia [10]. China is one of the countries rich in licorice. It is mainly found in the arid and semi-arid regions of north-eastern, northern, and north-western China. Xinjiang is the most common region [11]. Cui et al. (2023) [12] reported that licorice could prevent soil pollution and desertification, reduce soil erosion, and protect the ecological environment. The wild licorice resources of Xinjiang rank first in China, with southern Xinjiang accounting for over 70%. Licorice, with abundant protein, glycyrrhizin, flavonoids, and polysaccharides, has been a promising source of green fodder for animal feeding [13]. Abarghuei et al. (2021) [14] reported that the aerial parts of Glycyrrhiza are high-quality feeds for cattle and sheep. During the process of preparing hay, licorice stems and leaves fall off severely, resulting in severe nutrient loss. However, making a single licorice silage is not easy to achieve [15]. Therefore, mixing sweet sorghum with licorice stems and leaves to make silage can overcome their respective shortcomings. Liu et al. (2023) [16] reported that mixed silage with different proportions of alfalfa and maize can improve the nutritional value of silage. Ni et al. (2018) [17] found that forage soybean ensiled with corn or sorghum could be an alternative approach to improve forage soybean silage quality. Recently, some studies found that mixed silages with different forages could be a feasible way to improve the silage quality and aerobic stability of the fermentation system compared with the sole fermentation of various forages [18,19]. In conclusion, a mixed silage of legumes and gramineous plants has a better success rate than single-legume silage, and the mixed silage has improved fermentation quality, nutritional value, digestion, and metabolism for ruminant feeding. And the quality of mixed silage is closely related to the mixing ratio.

Recent studies have mainly reported the use of SS and alfalfa mixed silage in the total mixing ratio [20], the bioaugmentation effect of rumen fluid on SS silage [21], the nutritional value and fermentation characteristics of silage of different SS varieties [22], or different ratios of SS silage and corn silage in lactating dairy cows [23]. This study hypothesized that by using appropriate proportions, mixed silage with SS and LC might provide comprehensive nutrition and could also improve the fermentation quality of mixed silages. Therefore, this study examined the fermentation quality, aerobic stability, in vitro digestibility, and rumen degradation characteristics of silages mixed with sweet sorghum and aerial parts of licorice. We expected that the results obtained in this research could provide useful information for the practical use of mixed silages to feed ruminants.

## 2. Materials and Methods

### 2.1. Silage Mixture Preparation

The Cowley variety of SS with a 22.5% Brix value and wild LC were collected at the Agricultural Research Station of Tarim University, Xinjiang, China (longitude 81°31′ E, latitude 40°56′ N). The whole plant of SS at the maturity stage and aerial parts of licorice at the setting stage (50% setting rate) were harvested and chopped into 2–3 cm particle sizes by a multi-functional chopper (9DF53, Yanbei Animal Husbandry Machinery Group Co. Ltd., Beijing, China). A silage wrapping machine (D5552, Qufu Tianliang Trading Co. Ltd., Shandong, China) was used for making five types of SS-LC silage mixtures with different SS-to-LC ratios: (i) 0%SS + 100%LC (0%SS); (ii) 25%SS + 75%LC (25%SS); (iii) 50%SS + 50%LC (50%SS); (iv) 75%SS + 25%LC (75%SS); (v) 100%SS + 0%LC (100%SS). After careful blending, 5.5 kg of each mixture was placed in a 10 L lab hopper (polyethylene

flask fitted with an air-tight topper and sealed with screwcaps and plastic bands). Each processing group had 6 replicates. This anaerobic fermentation process was conducted at room temperature (20–25 °C) for 150 days.

### 2.2. Chemical and Microbial Analyses of Mixed Silages with Sweet Sorghum and Licorice at Different Proportions of Sweet Sorghum and Licorice

At the time of sampling, the pre-ensiled samples (fresh ingredients) were divided into two sub-samples, and each ensiled sample was divided into three sub-samples. The first sub-sample was oven-dried at 60 °C for 48 h to determine the dry matter (DM) content. The obtained powder samples were stored for later analysis of crude ash (Ash), acid detergent fiber (ADF), ether extract (EE), neutral detergent fiber (NDF), total nitrogen (TN), and water-soluble carbohydrates (WSCs) [24]. TN $\times$ 6.25 was used to calculate crude protein (CP). After reaction with an anthrone reagent [25], WSC content was determined by colorimetry. Van Soest et al.'s (1991) procedures were used to determine ADF and NDF contents [26].

A 20 g sample of silage mixture from each treatment group was weighed separately, thoroughly mixed with 180 mL of distilled water, and left for 24 h at 4 °C. Two layers of gauze and filter paper were then used to filter the extract samples. The filtrate was used to determine ammonia nitrogen ($NH_3$-N), organic acid, and pH content. The pH was measured with a HANNAHI2221 pH-measuring instrument. The supernatant was centrifuged at 10,000$\times$ $g$ for 10 min and collected for $NH_3$-N analysis [27].

After opening the lid for sampling and mixing, a sample of about 20 g was mixed with 180 mL distilled water in a 250 mL conical bottle, which was sealed before placing it into a refrigerator at 4 °C for 24 h. The suspension was then filtered first with four layers of gauze followed by a quantitative filter paper, and the filtered liquid was stored for later use. The extracts were centrifuged at 1500 r/min and mixed with 25% metaphoric acid solution at 5:1. The acidified supernatants were then loaded on a Thermo Scientific UltiMate 3000 high-performance liquid chromatograph (UltiMate XB-C18 column; Column temperature: 35 °C; Mobile phase: 0.1 mol/L potassium dihydrogen phosphate ($KH_2PO_4$); Flow rate: 1 mL/min) to determine the contents of lactic acid (LA), acetic acid (AA), propionic acid (PA), and butyric acid (BA) [28].

Representative and replicated silage samples were extracted on 0, 5, 10, 15, 20, and 25 days of aerobic exposure after successful ensiling. Each extract was diluted to $10^{-1}$–$10^{-8}$ g/mL by multiple dilution methods. About 1 mL of each dilution was taken from different dilution ratios and coated on different media including potato glucose agar (PDA) medium, high-salt Chach culture medium, MRS Medium, and ordinary agar medium. After incubating at 37 °C, 28 °C, 28 °C, and 28 °C for 48 h, yeast, mold, lactic acid bacteria (LAB), and aerobic bacteria (AB) were counted, respectively. The identification of the colonies was then carried out using special media, colony characterization, and microbiological microscopy. There were 3 replicates for each dilution ratio. The average colony count multiplied by the dilution of the appropriate dilution ratio was calculated as the number of microorganisms for the exposure time (CFU/g FM), and the results were expressed as log(CFU/g FM) [29,30].

The V-score method was used to evaluate the silage quality [31]. The V-score was calculated on a 100-point scale as follows: <60 (poor), 60–80 (fair), and 80–100 (good). The values of lactic acid/total acid, acetic acid/total acid, butyric acid/total acid, and $NH_3$-N/total nitrogen were comprehensively evaluated. The total score of organic acid was 100 points, the ratio of $NH_3$-N to total nitrogen was 50 points, and the comprehensive evaluation score was =(organic acid score)/2 + ($NH_3$-N ratio score of total nitrogen).

### 2.3. Aerobic Stability Analysis

A total of 180 silos (5 treatments $\times$ 6 exposure days $\times$ 6 replicates per treatment) were used for the aerobic stability test after 150 days of ensiling. Briefly, the mixed silages were removed from each silo, completely mixed, and loosely placed into a larger 10-L open-top polyethylene bottle. To prevent contamination by impurities such as fruit flies,

the bottles were stored at room temperature (20–25 °C) and wrapped in a double layer of gauze. Multi-channel temperature recorders (MDL-1048A high-precision temperature recorder, Shanghai Tianhe Automation Instrument Co., Ltd., Shanghai, China.) were used to measure temperature changes by placing their sensors in a central position. Six probes were placed as blanks in the environment. Temperatures were recorded every hour, with aerobic stability defined as the time required for the sample to rise 2 °C above room temperature [32]. After 0, 5, 10, 15, 20, and 25 days of aerobic exposure, the dynamics of microbe counts, $NH_3$-N, organic acids, and WSCs in the samples were analyzed. The determination method was the same as above.

### 2.4. In Vitro Incubation

The research undertaken complies with the current animal welfare laws in China. The experiment was carried out at the Animal Research Station of Tarim University, Xinjiang, China. All procedures used in this study were performed according to the Guidelines for the Care and Use of Animals for Research in China (GB 14925-2001).

Rumen fluid was collected from the rumens of three rumen-cannulated male *Duolang* sheep prior to feeding in the morning. The rumen fluid was immediately filtered using four layers of medical gauze, transferred to the laboratory, and stored in a water bath at 39 °C. The ingredient and nutrient compositions of the base diet (air-dried foundation) for sheep are shown in Table 1. Rumen fluid was immediately filtered, moved to the laboratory, and stored at 39 °C in a water bath. Prior to use, the rumen fluid was mixed with a 1:2 (rumen fluid:artificial rumen fluid) buffer solution, as described by Menke and Steingass (1988) [33]. The whole process of preparing the buffered rumen fluid was carried out under continuous flushing with $CO_2$.

**Table 1.** Ingredient and nutrient composition of the basal diet (dry matter foundation).

| Items | Content |
| --- | --- |
| composition of raw material (% DM [c]) | |
| Alfalfa | 25.00 |
| Maize straw | 25.00 |
| Corn | 29.37 |
| Soybean meal | 2.84 |
| Expanded soybean | 2.84 |
| Wheat bran | 4.27 |
| Cottonpulp | 1.89 |
| Spouting corn husks | 2.37 |
| Distillers Dried Grains with Solubles | 1.42 |
| Premix [a] | 5.00 |
| Total | 100.00 |
| Nutrient composition [b] (% FM [c]) | |
| Dry matter | 92.56 |
| Crude protein | 13.33 |
| Neutral detergent fiber | 38.82 |
| Acid detergent fiber | 22.32 |
| Ca | 0.65 |
| P | 0.32 |

[a] Premix is provided for each kilogram of diet: vitamin A 15,000 IU; vitamin D 5000 IU; vitamin E 50 mg; iron 90 mg; copper 12.5 mg; manganese 50 mg; zinc 100 mg; selenium 0.3 mg; iodine 0.8 mg; cobalt 0.5 mg. [b] Nutritional components were measured values. [c] FM, fresh matter; DM, dry matter.

In vitro fermentation was performed in serum flasks according to the method by Contreras-Govea et al. (2011) [34] with some modifications. Gas production, modeling of gas production, and assaying of relevant parameters of gas production refer to Ørskov et al. (1979) [35]. The equations described by Menke et al. (1979) [36] were used to calculate the metabolizable energy (ME) and digestible organic matter (DOM).

*2.5. The Degradation Characteristics of a Mixture Consisting of Sweet Sorghum and Licorice Stem and Leaf Silage in the Rumen of Sheep Were Evaluated Using the Nylon Bag Method*

The experiment was conducted using six Duolang sheep with permanent fistulae. For each group of test feed, six parallel samples were established and inserted into the rumen of each of the six sheep. Each parallel group consisted of a 5.00 g portion of test feed placed in a nylon bag of known weight. One replicate was used for each parallel group. The nylon bags, made from 300-mesh nylon sieve silk with double stitching, measured 50 mm × 100 mm. They were introduced into the rumen through the sheep's rumen fistula, and after 4, 8, 12, 24, 48, and 72 h of digestion, six parallel samples were simultaneously retrieved. The samples were rinsed with tap water and dried at 65 °C, and then undamaged degradation residues from each nylon bag were transferred to sample vials for testing. The Duolang sheep were fed and managed according to conventional practices, and the composition and nutrient content of the basal diet can be found in Table 1.

The rumen degradation rate and degradation parameters of a component of the feed to be tested at a given time point were determined with reference to Mirzaci-Aghsaghali et al. (2008) [37] and Sehu et al. (2010) [38];

$$A = (B - C)/B \times 100\% \tag{1}$$

Here, A is the rumen degradation rate (%) of an ingredient of the feed to be tested at a given point in time; B is the mass of an ingredient of the feed to be tested (%); and C is the mass of an ingredient in the residue (%).

Degradation rate curve:

$$P = a + b(1 - e^{-ct}) \tag{2}$$

Here, P is the rumen degradation rate (%) of the feed to be tested at a given time point; a is the fast-degradation fraction (%); b is the slow-degradation fraction (%); c is the degradation rate constant for b (%/h); and t is the incubation time point of the sample in the rumen (h) [35].

Effective rate of degradation:

$$ED = a + (b \times c)/(c + k) \tag{3}$$

Here, k is the rumen chyme efflux rate, taken as k = 0.031%/h.

Digestible energy (DE), metabolizable energy (ME), and net energy for gain (NEg) metrics in mixed-silage diets were estimated using prediction equations for the energy value of sheep (kJ/kg) [36].

Predictive modeling of DE:

$$DE = 19.509 - 0.170 \times NDF - 0.006 \times OM - 0.097 \times CP\left(R^2 = 0.973, p < 0.001\right) \tag{4}$$

Predictive modeling of ME:

$$ME = 0.046 + 0.820 \times DE\left(R^2 = 0.972, p < 0.001\right) \tag{5}$$

Predictive modeling of NEg:

$$NEg = ME \times 0.4571 \tag{6}$$

*2.6. Statistical Analysis*

Analysis of variance (ANOVA) was performed using the general linear model procedure of SPSS 26.0. A one-way ANOVA was performed on aerobic stability, chemical composition, fermentation quality, and in vitro incubation data. A two-way ANOVA was performed on chemical composition and microbial counts after aerobic exposure. The

statistical difference between the mean values was determined using Tukey's multiple comparisons test and was considered to be significant at $p < 0.05$.

## 3. Results

### 3.1. Chemical and Microbial Compositions of Pre-Ensiled and Ensiled Materials of the Mixed Silage of Sweet Sorghum and Aerial Parts of Licorice

The chemical composition of the mixed silage ingredients is shown in Table 2. Among all roughages, SS had a lower DM, CP, NDF, ADF, Ash, and ADL content and a higher WSC content. The content of EE was similar between SS and LC. After fermenting mixed silages for 150 days, the DM, CP, NDF, ADF, Ash, and ADL content decreased with the increasing level of SS silages (Table 3, $p < 0.05$). However, the WSC content increased in all mixed silages, which was between 3.60 and 8.48 (% DM). The 100%SS and 75%SS had significantly higher WSC content than that of 0%SS ($p < 0.05$). The NDF and ADF contents of all mixed silages were between 40.96 and 44.88 (% DM) and between 20.85 and 26.81 (% DM), respectively. The 0%SS and 25%SS groups were significantly higher than those in the other three groups ($p < 0.05$), but there was no significant difference between the 0%SS and 25%SS groups ($p > 0.05$). There were no significant differences in EE contents among all the silage mixtures ($p > 0.05$). The LAB population in all silage mixtures was greater than $10^5$ cfu/g FM, even if the content of SS in the SS-LC mixed silages was increased ($p < 0.05$). No significant difference was observed in the aerobic bacteria and yeast population among the five silage mixtures ($p > 0.05$), but lactic acid bacteria showed a significant upward trend.

**Table 2.** Chemical composition of ingredients used for the mixed silage of sweet sorghum and aerial parts of licorice.

| Items | Treatments | |
|---|---|---|
| | Sweet Sorghum | Licorice |
| Dry matter (% FM) | 33.78 | 37.15 |
| Crude protein (% DM) | 7.14 | 13.75 |
| Neutral detergent fiber (% DM) | 39.18 | 47.59 |
| Acid detergent fiber (% DM) | 19.24 | 31.64 |
| Crude ash (% DM) | 7.01 | 13.98 |
| Acid detergent lignin (% DM) | 5.51 | 10.54 |
| Water soluble carbohydrate (% DM) | 18.63 | 7.86 |
| Ether extract (% DM) | 2.13 | 2.14 |

**Table 3.** Effect of chemical compositions of the mixed silage of sweet sorghum and aerial parts of licorice after 150 days of ensiling.

| Items | Treatments | | | | | Mean | SEM [a] | *p*-Value |
|---|---|---|---|---|---|---|---|---|
| | 0%SS | 25%SS | 50%SS | 75%SS | 100%SS | | | |
| Chemical composition | | | | | | | | |
| Dry matter, DM (% FM) | 39.76 [A] | 37.91 [A] | 35.90 [B] | 35.11 [B] | 34.24 [B] | 36.59 | 0.54 | <0.001 |
| Crude protein, CP (% DM) | 14.17 [A] | 12.68 [A] | 9.35 [B] | 7.70 [B] | 6.49 [C] | 10.08 | 0.78 | <0.001 |
| Neutral detergent fiber, NDF (% DM) | 44.88 [A] | 43.88 [A] | 41.20 [B] | 42.82 [B] | 40.96 [C] | 42.75 | 0.40 | <0.001 |
| Acid detergent fiber, ADF (% DM) | 26.81 [A] | 27.37 [A] | 23.52 [B] | 22.93 [B] | 20.85 [C] | 24.03 | 0.66 | <0.001 |
| Crude ash, Ash (% DM) | 12.74 [A] | 12.18 [B] | 10.64 [C] | 9.43 [D] | 6.72 [E] | 10.34 | 0.58 | <0.001 |
| Acid detergent lignin, ADL (% DM) | 10.33 [A] | 9.17 [A] | 8.87 [B] | 7.74 [CD] | 5.01 [D] | 8.22 | 0.48 | <0.001 |
| Water soluble carbohydrate, WSC (% DM) | 3.60 [C] | 4.95 [B] | 5.72 [B] | 7.34 [A] | 8.48 [A] | 6.02 | 0.46 | <0.001 |
| Ether extract, EE (% DM) | 2.12 | 2.11 | 2.13 | 2.12 | 2.12 | 2.12 | 0.01 | 0.996 |
| Microbial composition (Log10 cfu/g FM [b]) | | | | | | | | |
| Lactic acid bacteria | 6.64 [C] | 7.12 [B] | 7.52 [B] | 8.21 [A] | 8.52 [A] | 7.60 | 0.19 | <0.001 |
| Aerobic bacteria | 5.42 | 5.39 | 5.46 | 5.33 | 5.43 | 5.41 | 0.02 | 0.350 |
| Yeasts | 3.08 | 3.15 | 3.11 | 3.05 | 3.01 | 3.08 | 0.02 | 0.099 |

[A–E] The same letter indicates non-significant differences ($p > 0.05$) and different letters indicate significant differences ($p < 0.05$). Capital letters are significant differences between treatment groups. [a] SEM, standard error of means. [b] cfu, colony-forming units.

### 3.2. Fermentation Characteristics of the Mixed Silages of Sweet Sorghum and Aerial Parts of Licorice

Table 4 shows that all mixed silages and their interactions had a significant effect on AA, LA, $NH_3$-N, and pH contents ($p < 0.05$). With the increasing level of SS, the LA content continuously increased, and the pH and $NH_3$-N gradually decreased after ensiling for 150 days. Although the content of SS in silage increased the LA content, it decreased the pH value and the $NH_3$-N, and AA content. The 50%SS, 75%SS, and 100%SS silages were better preserved than 0%SS and 25%SS silages on the basis of the V-score (Table 4). There was a fluctuating upward trend in the ratio of lactic acid to acetic acid (LA/AA). The SS-LC mixed silages, particularly 0%SS and 25%SS, had significantly ($p < 0.05$) lower LA/AA values than that of the 100%SS mixed silage. With the increasing level of the SS ratio, the content of BA in all mixed silages showed a downward trend, and the content of BA of 0%SS and 25%SS was significantly ($p < 0.05$) higher than those of the 50%SS, 75%SS, and 100%SS. However, there were no significant ($p > 0.05$) differences in PA content among all silage mixtures. With the increasing level of the SS ratio, the V-score value continuously increased after ensiling for 150 days. The V-score values of 50%SS, 75%SS, and 100%SS were 82.00, 85.50, and 88.00, respectively, which were all higher than 80 points. Therefore, the 50%SS, 75%SS, and 100%SS silages had high fermentation quality.

**Table 4.** Effect of fermentation characteristics of the mixed silage of sweet sorghum and aerial parts of licorice.

| Items | Treatments | | | | | Mean | SEM [a] | *p*-Value |
|---|---|---|---|---|---|---|---|---|
| | 0%SS | 25%SS | 50%SS | 75%SS | 100%SS | | | |
| pH | 5.01 [A] | 4.34 [B] | 4.15 [B] | 4.05 [B] | 4.01 [C] | 4.31 | 0.10 | <0.001 |
| Ammonia nitrogen (% TN [b]) | 4.85 [A] | 4.61 [AB] | 3.97 [B] | 3.86 [B] | 3.54 [B] | 4.17 | 0.13 | <0.001 |
| Lactic acid (% FM) | 1.42 [C] | 1.89 [BC] | 2.56 [B] | 3.14 [A] | 3.79 [A] | 2.56 | 0.23 | <0.001 |
| Acetic acid (% FM) | 1.16 [A] | 0.96 [B] | 0.85 [B] | 0.67 [C] | 0.56 [D] | 0.84 | 0.06 | 0.003 |
| Propanoic acid (% FM) | 0.08 | 0.08 | 0.08 | 0.09 | 0.08 | 0.08 | 0.01 | 0.782 |
| Butyric acid (% FM) | 0.18 [A] | 0.10 [A] | 0.06 [B] | 0.05 [B] | 0.03 [C] | 0.08 | 0.06 | <0.001 |
| Lactic acid/total acid | 34.23 [E] | 40.75 [D] | 45.07 [C] | 52.36 [B] | 54.89 [A] | 45.46 | 2.02 | <0.001 |
| Acetic acid/total acid | 28.72 [A] | 25.16 [B] | 22.36 [C] | 20.34 [D] | 19.05 [E] | 23.13 | 0.93 | <0.001 |
| Butyric acid/total acid | 16.56 [A] | 15.27 [B] | 10.35 [C] | 9.45 [D] | 6.74 [E] | 10.58 | 1.06 | <0.001 |
| V-score | 75.00 | 78.50 | 82.00 | 85.50 | 88.00 | - | - | - |
| Grade | fair | fair | good | good | good | - | - | - |

[A–E] The same letter indicates non-significant differences ($p > 0.05$) and different letters indicate significant differences ($p < 0.05$). [a] SEM, standard error of means. [b] TN, total nitrogen.

### 3.3. The Chemical Compositions of the Mixed Silage during Aerobic Exposure of the Mixed Silage of Sweet Sorghum and Aerial Parts of Licorice

As shown in Table 5, among all the treatment groups, with the increase in the SS ratio, the content of DM, pH, and AA showed a decreasing trend, which was significant in all treatment groups ($p < 0.05$). However, the content of WSC and LA showed an increasing trend, which was significant in all treatment groups ($p < 0.05$). With the increase in aerobic exposure time, the content of DM and pH showed an upward trend. However, there was a downward trend in AA, LA, and WSC contents. The relevant data of each aerobic exposure showed different degrees of significance ($p < 0.05$).

**Table 5.** Effect of aerobic exposure days on fermentative characteristics of the mixed silage of sweet sorghum and aerial parts of licorice.

| Items | Treatments | Aerobic Exposure Days | | | | | | Mean | SEM [a] | p-Value [b] | | |
|---|---|---|---|---|---|---|---|---|---|---|---|---|
| | | 0 | 5 | 10 | 15 | 20 | 25 | | | M | N | M × N |
| Dry matter (% FM) | 0%SS | 39.76 $^{Ca}$ | 42.09 $^{Ba}$ | 44.76 $^{Ba}$ | 46.68 $^{Ba}$ | 50.53 $^{Aa}$ | 53.62 $^{Aa}$ | 46.24 | 1.24 | <0.001 | <0.001 | <0.001 |
| | 25%SS | 37.91 $^{Cab}$ | 40.37 $^{Ba}$ | 42.23 $^{Bb}$ | 43.85 $^{Bb}$ | 45.01 $^{Ab}$ | 52.80 $^{Aa}$ | 43.70 | | | | |
| | 50%SS | 35.90 $^{Cbc}$ | 38.55 $^{Bb}$ | 40.81 $^{Bc}$ | 42.50 $^{Bbc}$ | 43.31 $^{Ac}$ | 48.24 $^{Ab}$ | 41.55 | | | | |
| | 75%SS | 35.11 $^{Cc}$ | 38.33 $^{Bb}$ | 40.18 $^{Bc}$ | 41.77 $^{Bc}$ | 43.26 $^{Ac}$ | 45.95 $^{Ac}$ | 40.77 | | | | |
| | 100%SS | 34.24 $^{Cc}$ | 36.53 $^{Bc}$ | 37.74 $^{Bd}$ | 38.93 $^{Bd}$ | 39.35 $^{Bd}$ | 43.02 $^{Ad}$ | 38.30 | | | | |
| Water soluble carbohydrate (% FM) | 0%SS | 3.60 $^{Ac}$ | 2.38 $^{Ad}$ | 2.01 $^{Bd}$ | 1.79 $^{B}$c | 1.68 $^{Ce}$ | 1.53 $^{Cd}$ | 2.17 | 0.34 | 0.028 | <0.001 | <0.001 |
| | 25%SS | 4.95 $^{Ab}$ | 4.26 $^{Ac}$ | 3.89 $^{Ac}$ | 3.15 $^{Bb}$ | 2.77 $^{Bd}$ | 2.25 $^{Cc}$ | 3.55 | | | | |
| | 50%SS | 5.72 $^{Ab}$ | 5.54 $^{Ac}$ | 5.12 $^{Bb}$ | 4.47 $^{Bb}$ | 3.99 $^{Cc}$ | 3.78 $^{Cb}$ | 4.77 | | | | |
| | 75%SS | 7.34 $^{Aa}$ | 6.79 $^{Ab}$ | 6.24 $^{Ba}$ | 5.62 $^{Ba}$ | 4.68 $^{Cb}$ | 4.16 $^{Cab}$ | 5.81 | | | | |
| | 100%SS | 8.48 $^{Aa}$ | 7.84 $^{Aa}$ | 7.16 $^{Aa}$ | 6.29 $^{Ba}$ | 5.62 $^{Ba}$ | 4.61 $^{Ca}$ | 6.67 | | | | |
| pH | 0%SS | 5.01 $^{Ca}$ | 6.56 $^{Ba}$ | 6.89 $^{Ba}$ | 7.17 $^{Aa}$ | 7.59 $^{Aa}$ | 7.83 $^{Aa}$ | 6.84 | 0.04 | <0.001 | <0.001 | <0.001 |
| | 25%SS | 4.34 $^{Cb}$ | 5.02 $^{Bb}$ | 5.75 $^{Bb}$ | 6.98 $^{Ab}$ | 7.34 $^{Ab}$ | 7.65 $^{Ab}$ | 6.18 | | | | |
| | 50%SS | 4.15 $^{Cb}$ | 4.68 $^{Bb}$ | 5.50 $^{Bb}$ | 6.72 $^{Abc}$ | 7.56 $^{Abc}$ | 7.75 $^{Ab}$ | 6.06 | | | | |
| | 75%SS | 4.05 $^{Cb}$ | 4.53 $^{Cbc}$ | 5.48 $^{Bbc}$ | 6.54 $^{Ab}$ | 7.48 $^{Ab}$ | 7.75 $^{Ab}$ | 5.97 | | | | |
| | 100%SS | 4.01 $^{Cc}$ | 4.34 $^{Cc}$ | 5.36 $^{Bc}$ | 6.51 $^{Ac}$ | 7.40 $^{Ac}$ | 7.74 $^{Ab}$ | 5.89 | | | | |
| Lactic acid (% FM) | 0%SS | 1.42 $^{Ac}$ | 1.12 $^{Ac}$ | 0.96 $^{Ad}$ | 0.83 $^{Bc}$ | 0.80 $^{Bc}$ | 0.76 $^{Bc}$ | 0.98 | 0.21 | <0.001 | <0.001 | <0.001 |
| | 25%SS | 1.89 $^{Abc}$ | 1.58 $^{Ab}$ | 1.32 $^{Ac}$ | 1.10 $^{Bc}$ | 1.05 $^{Bc}$ | 0.99 $^{Bc}$ | 1.32 | | | | |
| | 50%SS | 2.56 $^{Ab}$ | 2.28 $^{Ab}$ | 2.01 $^{Bb}$ | 1.85 $^{Bb}$ | 1.62 $^{Cb}$ | 1.34 $^{Cb}$ | 1.94 | | | | |
| | 75%SS | 3.14 $^{Aa}$ | 2.81 $^{Aa}$ | 2.53 $^{Ba}$ | 2.16 $^{Ba}$ | 1.96 $^{Cb}$ | 1.65 $^{Cb}$ | 2.38 | | | | |
| | 100%SS | 3.79 $^{Aa}$ | 3.24 $^{Aa}$ | 2.92 $^{Ba}$ | 2.74 $^{Ba}$ | 2.43 $^{Ba}$ | 2.10 $^{Ca}$ | 2.87 | | | | |
| Acetic acid (% FM) | 0%SS | 1.16 $^{Aa}$ | 0.89 $^{Aa}$ | 0.75 $^{Aa}$ | 0.63 $^{Ba}$ | 0.54 $^{Ba}$ | 0.41 $^{Ba}$ | 0.73 | 0.05 | <0.001 | <0.001 | <0.001 |
| | 25%SS | 0.96 $^{Ab}$ | 0.74 $^{Ab}$ | 0.62 $^{Ab}$ | 0.54 $^{Bb}$ | 0.42 $^{Bb}$ | 0.38 $^{Ba}$ | 0.61 | | | | |
| | 50%SS | 0.85 $^{Ab}$ | 0.62 $^{Ab}$ | 0.54 $^{Ab}$ | 0.43 $^{Bc}$ | 0.37 $^{Bb}$ | 0.32 $^{Bb}$ | 0.52 | | | | |
| | 75%SS | 0.67 $^{Ac}$ | 0.52 $^{Ac}$ | 0.41 $^{Ac}$ | 0.35 $^{Ad}$ | 0.30 $^{Bc}$ | 0.28 $^{Bb}$ | 0.42 | | | | |
| | 100%SS | 0.56 $^{Ad}$ | 0.47 $^{Ad}$ | 0.43 $^{Ac}$ | 0.36 $^{Ad}$ | 0.30 $^{Bc}$ | 0.27 $^{Bb}$ | 0.40 | | | | |

The same letter indicates non-significant differences ($p > 0.05$) and different letters indicate significant differences ($p < 0.05$). Uppercase letters show significance between different times. Lowercase letters show significance between different treatment groups. [a] SEM, standard error of means. [b] M, aerobic exposure days; N, treatments; M × N, the interaction between aerobic exposure days and treatments.

### 3.4. The Aerobic Stability of the Mixed Silage during Aerobic Exposure of the Mixed Silage of Sweet Sorghum and Aerial Parts of Licorice

Figure 1 shows the time taken for the temperature of mixed silage to rise by 2 °C above the ambient temperature after aerobic exposure. The 100%SS silage was the first to exceed 2 °C, followed by that of the 75%SS, 50%SS, 25%SS, and 0%SS. Figure 1 shows that the aerobic stability of 0%SS, 25%SS, 50%SS, 75%SS, and 100%SS silages decreased by 384 h, 360 h, 336 h, 288 h, and 240 h, respectively. The aerobic stability of 0%SS was significantly higher than that of the other four groups ($p < 0.05$), which were 25%SS, 50%SS, 75%SS, and 100%SS. There was no significant difference between the 25%SS and 50%SS silages ($p > 0.05$).

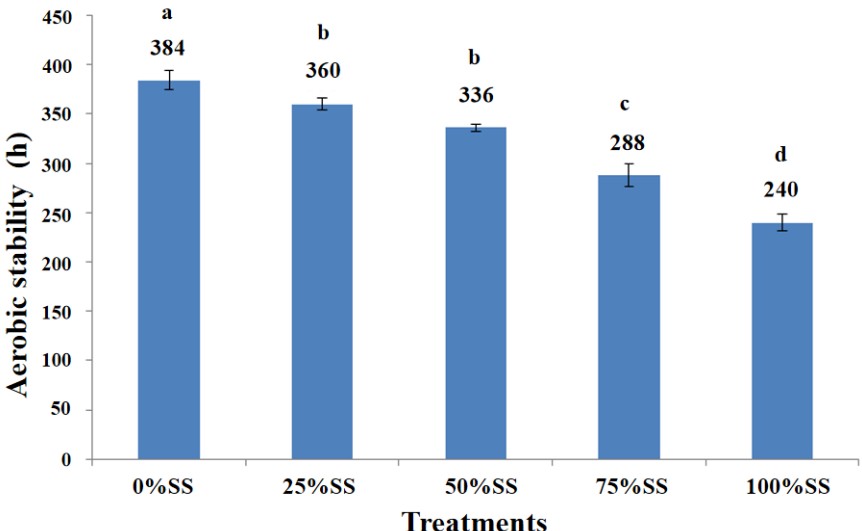

**Figure 1.** Aerobic stability of mixed silages during aerobic exposure ($n = 5$). The small black bars indicate the standard error of the mean. Lowercase letters show significance between different treatment groups.

### 3.5. The Microbial Changes of the Mixed Silage during Aerobic Exposure of the Mixed Silage of Sweet Sorghum and Aerial Parts of Licorice

As shown in Table 6, among all treatment groups, with the increase in the ratio of SS, the quantities of LAB, yeast, and AB showed an increasing trend, while the quantities of mold showed a decreasing trend. With the increase in aerobic exposure time, the number of LAB showed a decreasing trend because LAB is an anaerobic microorganism. The quantity of yeast, mold, and AB showed an increasing trend. Because they are aerobic microorganisms, the amount of air entering increases with an extension of aerobic exposure time.

### 3.6. Gas Production during In Vitro Fermentation of the Mixed Silages of Sweet Sorghum and Aerial Parts of Licorice

As shown in Figure 2, with an increase in in vitro fermentation time, the GP content of each treatment group increased gradually. The GP content of each treatment increased rapidly at 0–24 h. From 24 h to 48 h, the content of GP of each treatment group increased slowly until it stopped increasing at a later stage. Finally, the content of GP of each treatment group entered a plateau and was then relatively stable. The first plateau stage was seen in 0%SS followed by 25%SS, 75%SS, 50%SS, and 100%SS. At 72 h, the content of GP of 100%SS and 50%SS was significantly higher than that of 0%SS, 25%SS, and 75%SS. There was a clear difference between the 100% and 0% groups. However, the plateau of gas production for 100%SS occurred at 60 h and at 36 h for 0%SS.

**Table 6.** Effect of aerobic exposure days on microbial composition of the mixed silage of sweet sorghum and aerial parts of licorice.

| Items [b] | Treatments | Aerobic Exposure Days | | | | | | Mean | SEM [a] | p-Value | | |
|---|---|---|---|---|---|---|---|---|---|---|---|---|
| | | 0 | 5 | 10 | 15 | 20 | 25 | | | M | N | M × N |
| Lactic acid bacteria (Log$_{10}$ cfu/g FM) | 0%SS | 6.64 $^{Ac}$ | 6.13 $^{Ad}$ | 5.86 $^{Bc}$ | 5.35 $^{Bb}$ | 4.15 $^{Cc}$ | 3.65 $^{C}$ | 5.30 | 0.12 | <0.001 | <0.001 | <0.001 |
| | 25%SS | 7.12 $^{Ab}$ | 6.76 $^{Ac}$ | 5.97 $^{Bb}$ | 5.21 $^{Bb}$ | 4.49 $^{Cb}$ | 3.75 $^{C}$ | 5.55 | | | | |
| | 50%SS | 7.52 $^{Ab}$ | 7.02 $^{Ac}$ | 6.16 $^{Bb}$ | 5.64 $^{Ba}$ | 4.78 $^{Ca}$ | 3.85 $^{C}$ | 5.83 | | | | |
| | 75%SS | 8.21 $^{Aa}$ | 7.31 $^{Ab}$ | 6.34 $^{Ba}$ | 5.42 $^{Ba}$ | 4.61 $^{Ca}$ | 3.83 $^{C}$ | 5.95 | | | | |
| | 100%SS | 8.52 $^{Aa}$ | 7.64 $^{Aa}$ | 6.69 $^{Ba}$ | 5.05 $^{Bc}$ | 4.45 $^{Cb}$ | 3.81 $^{C}$ | 6.03 | | | | |
| Yeast (Log$_{10}$ cfu/g FM) | 0%SS | 3.08 $^{C}$ | 3.20 $^{Cc}$ | 3.86 $^{Bc}$ | 4.27 $^{Bc}$ | 4.88 $^{Ac}$ | 5.05 $^{Ac}$ | 4.06 | 0.16 | <0.001 | <0.001 | <0.001 |
| | 25%SS | 3.15 $^{C}$ | 3.32 $^{Cb}$ | 3.94 $^{Bc}$ | 4.35 $^{Bc}$ | 5.10 $^{Ac}$ | 5.84 $^{Ab}$ | 4.28 | | | | |
| | 50%SS | 3.11 $^{C}$ | 3.46 $^{Cb}$ | 4.01 $^{Bb}$ | 4.62 $^{Bb}$ | 5.89 $^{Ab}$ | 6.31 $^{Ab}$ | 4.57 | | | | |
| | 75%SS | 3.05 $^{C}$ | 3.58 $^{Cb}$ | 4.24 $^{Bb}$ | 5.60 $^{Bb}$ | 6.84 $^{Aa}$ | 7.24 $^{Aa}$ | 5.09 | | | | |
| | 100%SS | 3.01 $^{C}$ | 3.99 $^{Ca}$ | 4.74 $^{Ba}$ | 5.86 $^{Ba}$ | 7.14 $^{Aa}$ | 7.68 $^{Aa}$ | 5.40 | | | | |
| Mold (Log$_{10}$ cfu/g FM) | 0%SS | 0 | 1.54 $^{C}$ | 3.53 $^{Ba}$ | 4.89 $^{Ba}$ | 5.24 $^{Aa}$ | 6.01 $^{Aa}$ | 3.54 | 0.08 | <0.001 | <0.001 | <0.001 |
| | 25%SS | 0 | 1.23 $^{C}$ | 3.24 $^{Ba}$ | 3.95 $^{Ba}$ | 4.87 $^{Ab}$ | 5.81 $^{Aa}$ | 3.18 | | | | |
| | 50%SS | 0 | 0 | 3.11 $^{Bb}$ | 3.64 $^{Bb}$ | 4.48 $^{Ab}$ | 5.65 $^{Ab}$ | 2.81 | | | | |
| | 75%SS | 0 | 0 | 3.02 $^{Bb}$ | 3.24 $^{Bc}$ | 4.15 $^{Ac}$ | 5.24 $^{Ac}$ | 2.61 | | | | |
| | 100%SS | 0 | 0 | 2.94 $^{Bc}$ | 3.11 $^{Bc}$ | 3.99 $^{Bc}$ | 5.08 $^{Ac}$ | 2.52 | | | | |
| Aerobic bacteria (Log$_{10}$ cfu/g FM) | 0%SS | 5.42 $^{B}$ | 5.68 $^{Bc}$ | 5.96 $^{Ac}$ | 6.13 $^{Ac}$ | 6.24 $^{Ac}$ | 6.34 $^{Ac}$ | 5.96 | 0.10 | <0.001 | <0.001 | <0.001 |
| | 25%SS | 5.39 $^{B}$ | 5.95 $^{Bb}$ | 6.17 $^{Ab}$ | 6.34 $^{Ab}$ | 6.51 $^{Ab}$ | 6.64 $^{Ab}$ | 6.17 | | | | |
| | 50%SS | 5.46 $^{B}$ | 6.00 $^{Bb}$ | 6.34 $^{Ab}$ | 6.57 $^{Ab}$ | 6.68 $^{Ab}$ | 6.95 $^{Ab}$ | 6.33 | | | | |
| | 75%SS | 5.33 $^{C}$ | 6.24 $^{Ba}$ | 6.62 $^{Ba}$ | 6.89 $^{Aa}$ | 7.02 $^{Aa}$ | 7.22 $^{Aa}$ | 6.55 | | | | |
| | 100%SS | 5.43 $^{C}$ | 6.59 $^{Ba}$ | 7.01 $^{Aa}$ | 7.16 $^{Aa}$ | 7.30 $^{Aa}$ | 7.42 $^{Aa}$ | 6.82 | | | | |

The same letter indicates non-significant differences (*p* > 0.05) and different letters indicate significant differences (*p* < 0.05). Uppercase letters show significance between different times. Lowercase letters show significance between different treatment groups. [a] SEM, standard error of means. [b] M, aerobic exposure days; N, treatments; M × N, the interaction between aerobic exposure days and treatments.

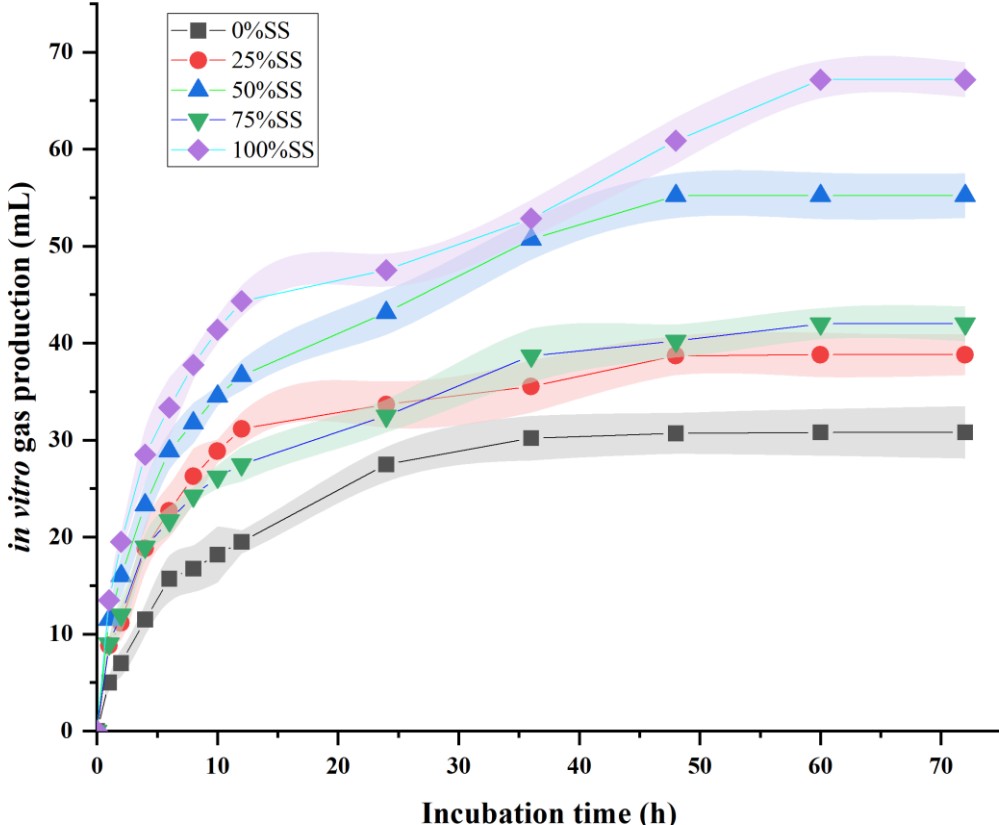

**Figure 2.** Gas production profiles (mL/g DM) from in vitro fermentation of the mixed silages for 72 h (*n* = 13). The width of the ribbon indicates the standard error.

*3.7. In Vitro Parameters of the 150-Day Mixed Silages of the Mixed Silage of Sweet Sorghum and Aerial Parts of Licorice*

As shown in Tables 7–10, there were significant differences in all index data among each treatment group. The content of potential GP, GP rate constant, DOM, ME, and IVDMD of 50%SS and 100%SS were significantly higher than those of the 0%SS, 25%SS, and 75%SS treatments (*p* < 0.05), and there was no significant difference between 50%SS and 100%SS (*p* > 0.05). Among all treatment groups, with the increasing level of the SS ratio, the pH showed a decreasing trend, ranging from 6.41 to 6.85. The content of $NH_3$-N in 50%SS and 100%SS was significantly lower than that in 0%SS, 25%SS, and 75%SS (*p* < 0.05). The content of $NH_3$-N in 50%SS and 100%SS was not significantly different (*p* > 0.05) and was 28.51 mg/dL and 25.32 mg/dL, respectively.

**Table 7.** Fermentation and gas production parameters from in vitro fermentation of TMR silage.

| Items [a] | Treatments | | | | | Mean | SEM | *p*-Value |
|---|---|---|---|---|---|---|---|---|
| | 0%SS | 25%SS | 50%SS | 75%SS | 100%SS | | | |
| potential GP (mL) | 33.78 [C] | 53.23 [B] | 62.42 [A] | 55.82 [B] | 66.96 [A] | 54.44 | 3.05 | <0.001 |
| GP rate constant (c, mL/h) | 0.06 [C] | 0.07 [B] | 0.09 [A] | 0.07 [B] | 0.10 [A] | 0.08 | 0.02 | <0.001 |
| pH | 6.85 [A] | 6.65 [B] | 6.55 [B] | 6.42 [C] | 6.41 [C] | 6.58 | 0.05 | <0.001 |
| $NH_3$-N (mg/dL) | 46.48 [A] | 36.68 [B] | 28.51 [C] | 33.17 [B] | 25.32 [C] | 34.03 | 1.96 | <0.001 |
| DOM (%) | 57.33 [C] | 62.06 [B] | 69.33 [A] | 61.16 [B] | 72.66 [A] | 64.51 | 1.50 | <0.001 |
| ME (MJ/kg DM) | 6.50 [C] | 7.47 [B] | 8.95 [A] | 7.28 [B] | 9.63 [A] | 7.79 | 0.31 | <0.001 |
| IVDMD (%) | 38.44 [C] | 55.18 [B] | 64.98 [A] | 56.98 [B] | 68.56 [A] | 56.83 | 2.79 | <0.001 |

[A–C] The same letter indicates non-significant differences (*p* > 0.05) and different letters indicate significant differences (*p* < 0.05). Capital letters are significant differences between treatment groups. [a] potential GP, potential gas production; GP rate constant, gas production rate constant; $NH_3$-N, ammonia nitrogen; DOM, digestible organic matter; ME, metabolizable energy; IVDMD, in vitro dry matter digestibility; SEM, standard error of means.

**Table 8.** The content of IVDMD of mixed silage with different ratios of SS-LC in vitro fermentation.

| Time/h | Treatments | | | | | Mean | SEM [a] | *p*-Value |
|---|---|---|---|---|---|---|---|---|
| | 0%SS | 25%SS | 50%SS | 75%SS | 100%SS | | | |
| 4 | 13.26 [C] | 14.55 [B] | 15.76 [A] | 15.21 [A] | 16.65 [A] | 15.08 | 0.79 | <0.001 |
| 8 | 15.54 [C] | 20.06 [B] | 23.42 [A] | 18.22 [B] | 25.35 [A] | 20.51 | 0.47 | <0.001 |
| 12 | 20.38 [C] | 25.17 [B] | 28.19 [A] | 26.43 [B] | 30.11 [A] | 26.05 | 0.54 | <0.001 |
| 24 | 28.46 [C] | 38.63 [B] | 47.63 [A] | 42.24 [B] | 50.65 [A] | 41.52 | 0.82 | <0.001 |
| 36 | 32.28 [C] | 45.52 [B] | 53.45 [A] | 48.61 [B] | 56.29 [A] | 47.23 | 0.74 | <0.001 |
| 48 | 35.16 [C] | 48.33 [B] | 56.27 [A] | 50.53 [B] | 61.42 [A] | 50.34 | 0.47 | <0.001 |
| 72 | 38.44 [C] | 55.18 [B] | 64.98 [A] | 56.98 [B] | 68.56 [A] | 56.83 | 2.79 | <0.001 |

[A–C] The same letter indicates non-significant differences (*p* > 0.05) and different letters indicate significant differences (*p* < 0.05). Capital letters are significant differences between treatment groups. [a] SEM, standard error of means.

**Table 9.** The content of pH of mixed silage with different ratios of SS-LC in vitro fermentation.

| Time/h | Treatments | | | | | Mean | SEM [a] | *p*-Value |
|---|---|---|---|---|---|---|---|---|
| | 0%SS | 25%SS | 50%SS | 75%SS | 100%SS | | | |
| 4 | 7.45 | 7.39 | 7.34 | 7.28 | 7.45 | 7.34 | 0.03 | <0.001 |
| 8 | 7.33 | 7.29 | 7.27 | 7.21 | 7.33 | 7.23 | 0.02 | <0.001 |
| 12 | 7.18 | 7.10 | 7.04 | 6.92 | 7.18 | 7.01 | 0.02 | <0.001 |
| 24 | 7.02 | 6.82 | 6.79 | 6.63 | 7.02 | 6.77 | 0.02 | <0.001 |
| 36 | 6.92 | 6.87 | 6.70 | 6.56 | 6.92 | 6.71 | 0.02 | <0.001 |
| 48 | 6.87 | 6.73 | 6.62 | 6.48 | 6.87 | 6.63 | 0.02 | <0.001 |
| 72 | 6.85 [A] | 6.65 [B] | 6.55 [B] | 6.42 [C] | 6.41 [C] | 6.58 | 0.05 | <0.001 |

[A–C] The same letter indicates non-significant differences (*p* > 0.05) and different letters indicate significant differences (*p* < 0.05). Capital letters are significant differences between treatment groups. [a] SEM, standard error of means.

**Table 10.** The content of NH$_3$-N of mixed silage with different ratios of SS-LC in vitro fermentation.

| Time/h | Treatments | | | | | Mean | SEM [a] | *p*-Value |
|---|---|---|---|---|---|---|---|---|
| | 0%SS | 25%SS | 50%SS | 75%SS | 100%SS | | | |
| 4 | 41.42 [A] | 32.08 [B] | 24.24 [C] | 28.63 [B] | 22.64 [C] | 29.80 | 0.64 | <0.001 |
| 8 | 43.38 [A] | 34.37 [B] | 26.16 [C] | 30.11 [B] | 24.52 [C] | 31.71 | 0.56 | <0.001 |
| 12 | 47.26 [A] | 36.61 [B] | 28.50 [C] | 33.26 [B] | 26.63 [C] | 34.45 | 0.65 | <0.001 |
| 24 | 50.16 [A] | 39.18 [B] | 31.61 [C] | 35.37 [B] | 28.48 [C] | 36.96 | 0.63 | <0.001 |
| 36 | 48.54 [A] | 38.34 [B] | 30.24 [C] | 34.16 [B] | 27.30 [C] | 35.72 | 0.44 | <0.001 |
| 48 | 47.34 [A] | 37.52 [B] | 29.11 [C] | 33.37 [B] | 26.45 [C] | 34.76 | 0.56 | <0.001 |
| 72 | 46.48 [A] | 36.68 [B] | 28.51 [C] | 33.17 [B] | 25.32 [C] | 34.03 | 1.96 | <0.001 |

[A–C] The same letter indicates non-significant differences (*p* > 0.05) and different letters indicate significant differences (*p* < 0.05). Capital letters are significant differences between treatment groups. [a] SEM, standard error of means.

*3.8. Degradation Rates of DM, CP, ADF, NDF, and ADL in the Rumen of Sheep of Mixed Silage of Sweet Sorghum and Licorice with Aerial Parts*

The results of Formulas (1)–(6) under subheading 2.5 of Materials and Methods are shown in Tables 11–16. The content of DE, ME, and Neg of 50%SS, 75%SS, and 100%SS were significantly higher than in 25%SS and 0%SS (*p* < 0.05). On the one hand, the rate of degradation of DM showed an increasing trend as the proportion of sweet sorghum increased and was significantly higher in 100%SS than in 0%SS (*p* < 0.05); the CP, NDF, ADF, and ADL contents were significantly higher in 25%SS and 50%SS than in 0%SS, 75%SS, and 100%SS (*p* < 0.05), and the difference was not significant in 25%SS and 50%SS (*p* > 0.05); the ADL content of 50%SS, 75%SS, and 100%SS was significantly higher than that of 0%SS and 25%SS (*p* < 0.05), and the difference between 50%SS, 75%SS, and 100%SS was not significant



(*p* > 0.05). On the other hand, with the increase in rumen degradation time, the rates of degradation of DM, ADF, NDF, ADL, and CP showed an increasing trend in each treatment group. In conclusion, 25%SS and 50%SS showed the best degradation rate.

**Table 11.** Energy value of mixed silage with different ratios of SS-LC (dry matter basis).

| Items/(kJ/kg) [a] | Treatments | | | | | Mean | SEM [a] | *p*-Value |
|---|---|---|---|---|---|---|---|---|
| | 0%SS | 25%SS | 50%SS | 75%SS | 100%SS | | | |
| DE [a] | 10.36 [B] | 10.67 [B] | 11.45 [A] | 11.33 [A] | 11.75 [A] | 11.11 | 0.48 | <0.001 |
| ME [a] | 8.54 [B] | 8.79 [B] | 9.43 [A] | 9.34 [A] | 9.68 [A] | 9.15 | 0.54 | <0.001 |
| Neg [a] | 3.90 [B] | 4.02 [B] | 4.31 [A] | 4.27 [A] | 4.42 [A] | 4.18 | 0.25 | <0.001 |

[A,B] The same letter indicates non-significant differences (*p* > 0.05) and different letters indicate significant differences (*p* < 0.05). Capital letters are significant differences between treatment groups. [a] DE, digestible energy; ME, metabolizable energy; NEg, net energy for gain; SEM, standard error of means.

**Table 12.** Degradation rate of the content of DM of mixed silage with different ratios of SS-LC in rumen of sheep.

| Items | Treatments | | | | | Mean | SEM | *p*-Value |
|---|---|---|---|---|---|---|---|---|
| | 0%SS | 25%SS | 50%SS | 75%SS | 100%SS | | | |
| DM | | | | | | | | |
| 4 (h) | 19.22 [B] | 23.42 [B] | 27.58 [A] | 29.11 [A] | 30.48 [A] | 25.96 | 0.13 | <0.001 |
| 8 (h) | 23.81 [C] | 27.02 [B] | 29.34 [B] | 30.01 [A] | 32.31 [A] | 28.49 | 0.13 | <0.001 |
| 12 (h) | 29.93 [C] | 33.31 [B] | 35.26 [B] | 37.24 [A] | 40.75 [A] | 35.29 | 0.14 | <0.001 |
| 24 (h) | 39.32 [C] | 44.05 [B] | 47.89 [B] | 50.37 [A] | 52.35 [A] | 46.79 | 0.15 | <0.001 |
| 48 (h) | 42.48 [C] | 48.22 [B] | 50.75 [B] | 54.71 [A] | 56.51 [A] | 50.53 | 0.15 | <0.001 |
| 72 (h) | 45.43 [C] | 53.35 [B] | 55.27 [B] | 58.33 [A] | 60.25 [A] | 54.52 | 0.27 | <0.001 |
| Degradability parameters of DM [a] | | | | | | | | |
| a (mL) | 10.30 [C] | 15.53 [B] | 18.98 [A] | 19.55 [A] | 20.62 [A] | 16.99 | 0.34 | <0.001 |
| b (mL) | 34.71 [C] | 37.63 [B] | 36.44 [B] | 39.54 [A] | 39.54 [A] | 37.57 | 0.53 | <0.001 |
| a + b (mL) | 45.01 [C] | 53.16 [B] | 55.42 [B] | 59.09 [A] | 60.16 [A] | 54.57 | 0.63 | <0.001 |
| c (mL/h) | 0.07 | 0.05 | 0.05 | 0.05 | 0.06 | 0.06 | 0.01 | <0.001 |
| ED (%) | 34.36 [C] | 38.76 [B] | 41.47 [B] | 43.96 [A] | 46.69 [A] | 41.05 | 0.41 | <0.001 |

[A–C] The same letter indicates non-significant differences (*p* > 0.05) and different letters indicate significant differences (*p* < 0.05). Capital letters are significant differences between treatment groups. [a] a, the rapidly degrading part; b, the slowly descending part; a + b, potentially degraded part; c, degradation rate constant; ED, effective degradation rate.

**Table 13.** Degradation rate of the content of CP of mixed silage with different ratios of SS-LC in rumen of sheep.

| Items | Treatments | | | | | Mean | SEM | *p*-Value |
|---|---|---|---|---|---|---|---|---|
| | 0%SS | 25%SS | 50%SS | 75%SS | 100%SS | | | |
| CP | | | | | | | | |
| 4 (h) | 30.62 [C] | 40.75 [A] | 37.53 [A] | 34.23 [B] | 28.25 [C] | 34.27 | 0.91 | <0.001 |
| 8 (h) | 33.02 [C] | 46.64 [A] | 43.12 [A] | 38.27 [B] | 30.38 [D] | 38.28 | 0.69 | <0.001 |
| 12 (h) | 36.34 [C] | 50.52 [A] | 46.35 [AB] | 42.16 [B] | 32.58 [C] | 41.59 | 0.86 | <0.001 |
| 24 (h) | 42.68 [B] | 53.21 [A] | 51.64 [A] | 45.35 [B] | 36.35 [C] | 45.84 | 0.66 | <0.001 |
| 48 (h) | 50.46 [C] | 56.53 [A] | 54.28 [A] | 52.16 [B] | 43.08 [D] | 51.30 | 0.52 | <0.001 |
| 72 (h) | 54.25 [B] | 62.21 [A] | 60.23 [A] | 58.34 [B] | 48.26 [C] | 56.65 | 0.48 | <0.001 |
| Degradability parameters of CP | | | | | | | | |
| a (mL) | 26.96 [C] | 38.15 [A] | 34.09 [A] | 32.94 [B] | 26.73 [C] | 31.77 | 0.36 | <0.001 |
| b (mL) | 31.10 [A] | 22.74 [C] | 25.00 [B] | 31.00 [A] | 32.85 [A] | 28.53 | 0.42 | <0.001 |

**Table 13.** *Cont.*

| Items | Treatments | | | | | Mean | SEM | *p*-Value |
|---|---|---|---|---|---|---|---|---|
| | 0%SS | 25%SS | 50%SS | 75%SS | 100%SS | | | |
| a + b (mL) | 58.06 C | 60.89 B | 59.09 B | 63.94 A | 59.58 B | 60.31 | 0.69 | <0.001 |
| c (mL/h) | 0.03 B | 0.05 A | 0.05 A | 0.02 B | 0.02 B | 0.03 | 0.01 | <0.001 |
| ED (%) | 42.26 B | 52.19 A | 49.52 A | 45.10 B | 39.6 C | 45.73 | 0.43 | <0.001 |

A–D The same letter indicates non-significant differences (*p* > 0.05) and different letters indicate significant differences (*p* < 0.05). Capital letters are significant differences between treatment groups. a, the rapidly degrading part; b, the slowly descending part; a + b, potentially degraded part; c, degradation rate constant; ED, effective degradation rate.

**Table 14.** Degradation rate of the content of NDF of mixed silage with different ratios of SS-LC in rumen of sheep.

| Items | Treatments | | | | | Mean | SEM | *p*-Value |
|---|---|---|---|---|---|---|---|---|
| | 0%SS | 25%SS | 50%SS | 75%SS | 100%SS | | | |
| NDF | | | | | | | | |
| 4 (h) | 22.16 B | 27.04 A | 25.35 A | 20.45 B | 18.15 C | 22.63 | 0.60 | <0.001 |
| 8 (h) | 27.24 B | 32.16 A | 30.24 A | 25.36 B | 23.23 C | 27.64 | 1.39 | <0.001 |
| 12 (h) | 35.04 B | 41.23 A | 39.34 A | 33.65 B | 32.42 C | 36.33 | 0.64 | <0.001 |
| 24 (h) | 47.85 B | 51.65 A | 50.22 A | 45.55 B | 42.04 C | 47.46 | 1.49 | <0.001 |
| 48 (h) | 60.24 B | 65.14 A | 63.16 A | 58.21 B | 56.21 C | 60.59 | 1.38 | <0.001 |
| 72 (h) | 64.34 B | 70.24 A | 67.34 A | 62.02 B | 60.37 C | 64.86 | 0.84 | <0.001 |
| Degradability parameters of NDF | | | | | | | | |
| a (mL) | 12.89 B | 18.80 A | 16.48 A | 11.34 B | 9.99 C | 13.90 | 0.72 | <0.001 |
| b (mL) | 54.1 | 54.45 | 53.42 | 53.32 | 53.51 | 53.76 | 0.83 | <0.001 |
| a + b (mL) | 66.99 B | 73.25 A | 69.90 A | 64.66 B | 63.50 B | 67.66 | 0.98 | <0.001 |
| c (mL/h) | 0.04 | 0.04 | 0.04 | 0.04 | 0.04 | 0.04 | 0.01 | <0.001 |
| ED (%) | 43.37 B | 49.48 A | 46.58 A | 41.38 B | 40.14 C | 44.19 | 1.172 | <0.001 |

A–C The same letter indicates non-significant differences (*p* > 0.05) and different letters indicate significant differences (*p* < 0.05). Capital letters are significant differences between treatment groups. a, the rapidly degrading part; b, the slowly descending part; a + b, potentially degraded part; c, degradation rate constant; ED, effective degradation rate.

**Table 15.** Degradation rate of the content of ADF of mixed silage with different ratios of SS-LC in rumen of sheep.

| Items | Treatments | | | | | Mean | SEM | *p*-Value |
|---|---|---|---|---|---|---|---|---|
| | 0%SS | 25%SS | 50%SS | 75%SS | 100%SS | | | |
| ADF | | | | | | | | |
| 4 (h) | 21.32 B | 25.25 A | 23.26 A | 20.34 B | 20.36 B | 22.10 | 0.43 | <0.001 |
| 8 (h) | 27.13 B | 30.34 A | 28.24 A | 26.15 B | 25.42 C | 27.45 | 0.42 | <0.001 |
| 12 (h) | 34.35 B | 37.22 A | 36.68 A | 33.22 BC | 32.51 C | 34.79 | 0.48 | <0.001 |
| 24 (h) | 44.02 B | 48.12 A | 46.12 A | 43.34 B | 41.25 C | 44.57 | 0.35 | <0.001 |
| 48 (h) | 53.16 B | 57.34 A | 55.22 A | 52.46 B | 50.55 C | 53.74 | 0.52 | <0.001 |
| 72 (h) | 63.31 B | 68.25 A | 66.12 A | 61.24 BC | 59.28 C | 63.64 | 0.42 | <0.001 |
| Degradability parameters of ADF | | | | | | | | |
| a (mL) | 16.20 B | 19.99 A | 17.99 A | 14.42 C | 15.31 B | 16.78 | 0.47 | <0.001 |
| b (mL) | 50.60 B | 53.31 A | 52.04 A | 49.18 B | 47.35 C | 50.49 | 0.37 | <0.001 |
| a + b (mL) | 66.80 B | 73.30 A | 70.04 A | 63.60 B | 62.66 C | 67.28 | 0.45 | <0.001 |
| c (mL/h) | 0.03 | 0.03 | 0.03 | 0.04 | 0.03 | 0.03 | 0.01 | <0.001 |
| ED (%) | 41.89 B | 45.76 A | 43.58 A | 40.85 B | 39.36 C | 42.28 | 0.64 | <0.001 |

A–C The same letter indicates non-significant differences (*p* > 0.05) and different letters indicate significant differences (*p* < 0.05). Capital letters are significant differences between treatment groups. a, the rapidly degrading part; b, the slowly descending part; a + b, potentially degraded part; c, degradation rate constant; ED, effective degradation rate.

**Table 16.** Degradation rate of the content of ADL of mixed silage with different ratios of SS-LC in rumen of sheep.

| Items | Treatments | | | | | Mean | SEM | *p*-Value |
|---|---|---|---|---|---|---|---|---|
| | 0%SS | 25%SS | 50%SS | 75%SS | 100%SS | | | |
| ADL | | | | | | | | |
| 4 (h) | 9.46 [C] | 10.09 [B] | 11.24 [A] | 11.85 [A] | 12.65 [A] | 11.05 | 0.84 | <0.001 |
| 8 (h) | 10.25 [C] | 12.02 [B] | 12.68 [B] | 12.34 [B] | 13.12 [A] | 12.08 | 0.93 | <0.001 |
| 12 (h) | 11.65 [D] | 13.11 [C] | 13.51 [B] | 13.26 [B] | 14.58 [A] | 13.22 | 0.93 | <0.001 |
| 24 (h) | 12.24 [C] | 14.84 [B] | 15.68 [A] | 15.37 [A] | 15.26 [A] | 14.67 | 0.93 | <0.001 |
| 48 (h) | 13.69 [C] | 15.31 [B] | 16.24 [A] | 16.08 [AB] | 15.59 [A] | 15.38 | 0.51 | <0.001 |
| 72 (h) | 14.22 [C] | 15.54 [B] | 16.56 [A] | 16.24 [A] | 16.61 [A] | 15.83 | 0.72 | <0.001 |
| Degradability parameters of ADL | | | | | | | | |
| a (mL) | 8.51 [B] | 7.29 [B] | 9.20 [A] | 10.24 [A] | 11.5 [A] | 9.34 | 0.56 | <0.001 |
| b (mL) | 5.75 [B] | 8.16 [A] | 7.34 [A] | 6.17 [B] | 4.76 [C] | 6.43 | 0.54 | <0.001 |
| a + b (mL) | 14.26 [B] | 15.45 [B] | 16.54 [A] | 16.41 [A] | 16.26 [A] | 15.78 | 1.09 | <0.001 |
| c (mL/h) | 0.05 [B] | 0.11 [A] | 0.08 [A] | 0.06 [B] | 0.07 [B] | 0.07 | 0.01 | <0.001 |
| ED (%) | 12.06 [C] | 13.66 [B] | 14.49 [A] | 14.31 [A] | 14.80 [A] | 13.86 | 0.96 | <0.001 |

[A–D] The same letter indicates non-significant differences ($p > 0.05$) and different letters indicate significant differences ($p < 0.05$). Capital letters are significant differences between treatment groups. a, the rapidly degrading part; b, the slowly descending part; a + b, potentially degraded part; c, degradation rate constant; ED, effective degradation rate.

## 4. Discussion

### 4.1. Chemical Compositions of SS-LC Mixed Silages

Except that the WSC contents of the ensiled silages were lower than their pre-ensiled counterparts, there were no obvious differences in the nutritional composition among the treatment groups. Similar results have been reported by Ni et al. (2018) [17], who found that the CP, EE, NDF, and ADF contents of ensiled silages were not significantly different from the pre-ensiled silages. Similar results have been reported by Ren et al. (2021) [21], who observed that the WSC contents in SS silage after 60 days were significantly lower than the pre-ensiled SS, and the WSC contents continued to reduce with increasing ensiling time.

After 150 days of ensiling, the CP, DM, Ash, and ADL contents in the mixed silages gradually decreased with the increasing amount of SS. While the WSC contents in mixed silages gradually increased, other nutrients did not change significantly. Interestingly, this is in agreement with Tássia et al. (2021) [39], Wang et al. (2020) [40], and Pedram et al. (2022) [41]. They reported reduced CP but increased WSC contents with increased proportions of gramineous plants in the mixed silages of legumes and gramineous plants.

The possible reason is that the grasses contain higher WSC but lower CP than legumes. The presence of the two forages in mixed silages helped to make the nutrition more comprehensive by improving the digestion nutritive value of the mixed silages [17,20,42].

### 4.2. Fermentation Characteristics of SS-LC Mixed Silages

Pre-ensiled forages with high WSC content (>5% DM) and sufficient LAB population (>$10^5$ cfu/g FM) were found in high-quality silage production by Weinberg et al. (2008) [43]. In this study, all mixtures of forages were able to be ensiled successfully. High-quality silage was reported to have high LA, low pH, low $NH_3$-N, and negligible BA contents by Catchpoole and Henzell (1971) [44]. Hence, as the proportion of SS increased, the fermentation quality of the mixed silages gradually improved. Interestingly, this is in agreement with Wang et al. (2021) [20], Zeng et al. (2020) [42], and Ni et al. (2017) [45], who reported that the fermentation quality of mixed silages would be gradually improved with the increase in the proportion of grasses and WSC content.

Dai et al. (2022) [46] found that the propagation of LAB was suppressed when the pH was below 4.00–4.20. Hence, the pH of mixed silages was below 4.20, which indicated that they tended to ensile successfully [46]. Auerbach and Nadeau (2020) [47] reported that better fermentation is indicated by a lower pH.

In this study, the pH of the 50%SS, 75%SS, and 100%SS was 4.15, 4.05, and 4.01, respectively. Therefore, 50%SS, 75%SS, and 100%SS were conducive to making high-quality silage because the 50%SS, 75%SS, and 100%SS silage had a sufficient LAB population, which produced a higher amount of lactic acid.

However, in all mixed silages, probably due to the low WSC and high DM contents, the pH gradually increased with increasing LC. Morgan et al. (1980) [48] reported that a high DM content may retard the proliferation of undesirable microorganisms and LAB in silage. The level of LC decreased the level of WSC in the mixed silages, which explains the reduction in LA, perhaps because there was less substrate available for LAB.

Tatulli et al. (2023) reported that when the WSC content was limited, heterofermentative LAB tended to be active, and several homofermentative strains such as *Lactobacillus plantarum* could carry out the lactic lactate/acetate conversion [49]. Keles and Demirci (2011) found that inoculation with a heterofermentative LAB (*Lactobacillus buchneri*) resulted in silage with higher ($p < 0.05$) concentrations of acetic and propionic acids and lower concentrations of LA and WSC. Therefore, in this study, the LC silage may be more inclined to be homofermentative, while the SS silage may be more inclined to be heterofermentative [50].

The $NH_3$-N content in silage is always indicative of protein breakdown and often defined as a product of amino acid deamination during silage fermentation [51,52]. In this study, as the proportion of LC increased, the $NH_3$-N content increased gradually. Similar results have been reported by Pedram et al. (2020) and Wu et al. (2022), who reported that when mixed silages were made of legume and Gramineae forages, the concentration of ammonia nitrogen increased gradually with an increase in the legume proportion in mixed forage-based silages [41,53].

### 4.3. Aerobic Exposure to LA and AA and Microbial Changes of SS-LC Mixed Silages

In tropical and subtropical regions, the high temperatures and humid climate during the summer months are major factors in the deterioration of silage mixes. Once a silo is opened, the aerobic microorganisms begin to multiply. Wilkinson and Davies (2012) showed good preservation of mixed silage, although this was subject to inherent instability following exposure to aerobes during the feed-out stage [32]. In fact, well-fermented silage is more unstable in the presence of oxygen than poorly fermented silage. Li et al. (2022) observed a negative correlation between the levels of lactic acid and WSC and the aerobic stability of silages [54]. In this study, similar observations showed that with an increase in the proportion of SS in mixed silages, the WSC and lactic acid contents in the mixed silages were gradually increased, and the aerobic stability was also improved. Johnson et al. (2002) showed that undesirable microbes can use LA and WSC as substrates to release carbon dioxide and heat, increasing the pH and increasing the nutrient loss when the silage is exposed to the air [55].

In this study, after aerobic exposure, with an increase in the SS proportion in mixed silages, the DM content gradually decreased, while aerobic bacteria and yeast gradually increased. As the ratio of SS increased, the number of aerobic bacteria and yeast increased, but the aerobic stability also gradually decreased. Interestingly, this is in agreement with Blajman et al. (2018) and Borreani et al. (2018), who reported that these samples were characterized by a lower DM that favored the activity of aerobic microorganisms, including yeasts, which, in the presence of air, depleted the available nutrient sources and contributed to the silage instability [56,57].

In this study, after 25 days of aerobic exposure, 50%SS, 75%SS, and 100%SS silages showed a strong increase in pH and a decrease in LA. The high levels of yeast, LA, and WSC in 50%SS, 75%SS, and 100%SS silages may explain this result. In general, yeasts are the main factor responsible for the aerobic deterioration of silage. When the yeast community exceeds $10^5$ cfu/g FM, silage deterioration tends to occur [58]. Aerobic bacteria were also found to be responsible for aerobic deterioration by Courtin and Spoelstra (1990) [59]. Similarly, after 150 days of aerobic exposure, more than $10^5$ cfu/g FM of yeasts and $10^7$ cfu/g FM of aerobic bacteria were found in 100%SS silage.

Acetic acid can effectively inhibit the proliferation of fungi, mold, and yeast during aerobic exposure and is known to be an important indicator for predicting the aerobic stability of silages [32]. Therefore, as reflected by the lower LAB and yeast populations, the 0%SS and 25%SS silages with higher AA contents were more stable during aerobic exposure than the control. Furthermore, the increased aerobic stability of the SS-LC mixture silages could be explained by the LC material used in this study.

### 4.4. In Vitro Parameters of SS-LC Mixed Silages

Digestibility has become widely accepted in the evaluation of the nutritional value and intake of feed [60]. In the meantime, the increase in the in vitro GP is commonly used as an indicator of the efficiency of the rumen digestibility and the predicted metabolizable energy of animal feed [34]. In this study, all GP and digestibility parameters, as well as the estimated ME, were improved by mixed silages of 50%SS and 100%SS. This is consistent with results from Bender et al. (2016) and Lucas et al. (2020) [61,62]. Mixed silage effectively improved digestibility and provided more complete nutrition. This indicates that SS-LC-based mixed silages do not have unfavorable effects on the rumen utilization of mixed silages, although no significant difference was observed between 75%SS and 25%SS silages.

### 4.5. Degradation Rates of DM, CP, ADF, NDF, and ADL in the Rumen of Sheep

The nylon bag method is a commonly used method for assessing the degradation characteristics of feeds and is suitable for studying the extent of feed degradation within the rumen of ruminants and its effect on rumen microbial activity. Cone et al. (2004) reported the efficiency of protein degradation and its effective utilization by employing a nylon pouch approach in the rumen and the use of Streptomyces staphylococcus protein-hydrolyzing enzyme preparations for the detection of rumen escape proteins in grass and grass silage [63]. The nylon bag method allows the rate of feed degradation to be assessed by monitoring the degree of degradation of the residue in the bag.

In this experiment, with an increase in the sweet sorghum ratio, the degradation rates of DE, ME, NEg, and ADL in the rumen of sheep showed an increasing trend, indicating that sweet sorghum was easier to digest, degrade, utilize, and absorb than licorice. Due to the high carbohydrate content in sweet sorghum, it can provide energy for rumen microorganisms. Ma et al. (2023) have shown that adding lactic acid bacteria additive to mixed silage of amaranth and corn straw can effectively improve the degradation of dry matter [64]. Microbial fermentation of sugars and organic acids in crops releases gases such as carbon dioxide and hydrogen, which leads to dry matter loss [57]. The results of this study are consistent with previous research indicating that an increase in sweet sorghum leads to a gradual increase in the rate of degradation of dry matter.

Lignin is deposited in the space between cellulose, hemicellulose, and pectin molecules in the secondary cell wall, where it forms a cross-linked network with other components of the cell wall [65,66]. This cross-linking helps to increase the strength and stiffness of the cell wall, allowing the plant to stand upright and resist external pressures [67]. Lignin is a complex structural polymer resistant to microbial degradation, and its deposition in plant cell walls makes the cellulose and hemicellulose components of cell walls less accessible to rumen microbial enzymes, thus reducing rumen fermentation efficiency and nutrient availability to animals [68]. In addition, lignin inhibits the activity of digestive enzymes in the small intestine, further reducing the digestibility of the feed [69]. The results are consistent with the present study. In this study, the ADL content of mixed silage gradually decreased as the proportion of sweet sorghum gradually increased, so that the mixed silage with lower lignification was more easily degraded. In terms of digestibility, the main difference between grasses and legumes is that while legumes with thicker lignified walls can only be lightly digested by rumen microbes, grass tissues with the same thick lignified walls degrade extensively, albeit slowly [70]. Therefore, in rumen degradation experiments, the ADL degradation rates of 75%SS and 100%SS were higher than those of other groups.

In this experiment, as the proportion of sweet sorghum increased, the degradation rates of CP, ADF, and NDF showed a trend of first increasing and then decreasing (Tables 13–15), indicating that the ratio of sweet sorghum and licorice mixed silage was the key factor, and only at the appropriate ratio could the effective components in the feed be effectively absorbed. At 25%SS and 50%SS, the bacterial community in the rumen is more conducive to the absorption of CP, ADF, and NDF nutrients, and a large number of fibrinolytic bacteria may be produced in the rumen of animals, such as *Ruminococcus albus*, *Fibrobacter succinogenes*, *Ruminococcus flavefaciens*, *Butyrivibrio fibrisolvens*, and *Bacillus*, etc. [71]. *Bacillus* can secrete many enzymes, such as amylase, xylanase, chitinase, β-1, 3-glucanase, β-glucosidase, lipase, protease, cellulase, etc. [72]. The rate of degradation of CP is influenced by the true protein concentration and amino acid composition of CP in the feed [73]. This result is consistent with that of the experiment. In this experiment, the crude protein content of 25%SS and 50%SS was relatively high, so that the rumen degradation rate of CP was high, which may be due to the large number of microorganisms that can secrete proteases during rumen degradation of the feed. However, although the crude protein content of 0%SS was the highest in this experiment, the degradation rate of the CP of 0%SS in the rumen was not the fastest, possibly because the feed composition of 0%SS was all leguminous licorice, there was no sweet sorghum in the gramineous family, and there was no material to provide carbohydrates for the microorganisms secreting protease to provide energy, which was not good for digestion and absorption.

Cellulase treatment breaks the connection between polyester and cellulose, which can be degraded and utilized by the microbiota in the rumen [74]. Rumen fibro-degrading bacteria degrade plant cellulose and hemicellulose mainly by producing a series of cellulases and hemicellulose enzymes. These enzymes, often referred to as cellulase complexes and hemi-cellulase complexes, are composed of a number of different enzymes, including β-glucosidase, xylanase, galacturonase, etc. [75]. These enzymes are able to break the chemical bonds between cellulose and hemicellulose, breaking it down into smaller soluble carbohydrates [76].

Rumen fibro-degrading bacteria produce volatile fatty acids (VFAs) in the process of degradation of cellulose and hemicellulose [77]. Volatile fatty acids include short-chain fatty acids such as acetic acid, propionic acid, and butyric acid, which are metabolites of cellulose and hemicellulose fermentation. These VFAs are absorbed by the animal as an energy source in the rumen and play an important role in the physiological function and nutritional status of the host animal [78]. However, the limitation of this paper is that the microorganisms and metabolites in the rumen fluid of different treatment groups have not been detected.

## 5. Conclusions

In conclusion, research was conducted on the fermentation quality, aerobic stability, degradation characteristics of nylon bags, and in vitro degradation characteristics of mixed silage of whole sweet sorghum and licorice with stems and leaves in different proportions, which showed that the mixed silage of whole sweet sorghum and licorice with stems and leaves in a 50:50 ratio (50%SS) effectively improved fermentation quality and rumen degradation rate. The addition of licorice with stems and leaves could effectively improve the aerobic stability and CP content of the mixed silages.

**Author Contributions:** Conceptualization, S.Z.; methodology, J.W. and F.C.; software, F.C.; validation, S.Z., J.W.; formal analysis, F.C.; investigation, J.W.; resources, F.C.; data curation, S.Z.; writing—original draft, F.C.; preparation, F.C.; writing—review and editing, J.W., A.S.C., H.K. and S.Z.; visualization, F.C.; supervision, S.Z. and J.W.; funding acquisition, S.Z. All authors have read and agreed to the published version of the manuscript.

**Funding:** This present work was funded by the National Natural Science Foundation of China (No. 32260858) and Key Laboratory of Tarim Animal Husbandry Science and Technology, Xinjiang Production & Construction Group (No. HS202101).

**Institutional Review Board Statement:** Not applicable.

**Informed Consent Statement:** Not applicable.

**Data Availability Statement:** All data are presented in this article in the form of figures and tables.

**Acknowledgments:** The authors also acknowledge the assistance of Instrumental Analysis Center of Tarim University.

**Conflicts of Interest:** The authors declare no conflicts of interest.

## Abbreviations

AA, acetic acid; AB, aerobic bacteria; ADF, acid detergent fiber; ADL, acid detergent lignin; Ash, crude ash; BA, butyric acid; CP, crude protein; DE, digestible energy; DM, dry matter; ED, effective degradation rate; EE, ether extract; FM, fresh matter; GP, gas production; IVDMD, in vitro dry matter digestibility; LA, high lactic acid; LAB, lactic acid bacteria; LC, aerial parts of licorice; ME, metabolizable energy; MOD, digestible organic matter; NDF, neutral detergent fiber; Neg, net energy for gain; $NH_3$-N, ammonia nitrogen; PA, propionic acid; SS, sweet sorghum; TN, total nitrogen; WSC, water-soluble carbohydrate.

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
