# Peer review of "Assessing Fermentation Quality, Aerobic Stability, In Vitro Digestibility, and Rumen Degradation Characteristics of Silages Mixed with Sweet Sorghum and Aerial Parts of Licorice"

_agriculture, doi:10.3390/agriculture14020212_

Round 1
Reviewer 1 Report
Comments and Suggestions for Authors
Authors need to provide procedure for determination of microbial contents in the study.

Comments on the Quality of English Language
The quality of English language is okay.
The authors should attend to some of the corrections made.
Author Response
Dear the Agriculture team,
We would like to thank you and the reviewers for taking time to review the present manuscript. The comments and suggestions are greatly appreciated, and the manuscript has been markedly improved after addressing the issues raised. All contents in the manuscript have been checked and revised carefully, and reviewers’ questions and suggestions have been responded point-by-point. Track revision mode was used for all changes in the revised manuscript.
Dr. Sujiang Zhang on behalf of all co-authors

Reviewer 2 Report
Comments and Suggestions for Authors
The paper gives an insight into various parameters during mixed fermentation of sorghum and licorice as animal feed. The use of these draught-tolerant species is interesting, especially in the climate change scenario.
It is well written, and below you have minor suggestions to improve readability
in the abstract try to simplify the description, maybe just what is between brackets?
"Five mixtures were produced on a dry matter (DM) basis: (i) 0%SS+100%LC (0%SS); (ii) 25%SS+75%LC (25%SS); (iii) 50%SS+50%LC (50%SS); (iv) 75%SS+25%LC (75%SS) and (v) 100%SS+0%LC (100%SS)"
Line 16-17 and 23-25: use, instead of ;
Line 33: avoid repeating area
Line 45: omit )
Line 92-93: the sentence is not clear, the temperature was 20 and 25 °C? use at room temperature of 20° C and 25 °C and omit respectively
Line 161: filtered how? the rumen fluid solution 1:1 was used as incubation media? Please describe briefly.
Line 235: not only 100 % but also 75% is significantly different
Line 243: why you do not report the statistical differences for LAB)
A general comment on the tabs 5 and 6. They would be more legible if there was a line between an item and the following
Line 324: the 50% reaches the plateau before the 75%. The figure 2 would be more comprehensible is the x-axes had the real incubation times
Line 368: do not use capital letters for with
Line 526-528: try to rephrase… this… this… is not clear
Line 617: include pages
Author Response
Dear the Agriculture team,
We would like to thank you and the reviewers for taking time to review the present manuscript. The comments and suggestions are greatly appreciated, and the manuscript has been markedly improved after addressing the issues raised. All contents in the manuscript have been checked and revised carefully, and reviewers’ questions and suggestions have been responded point-by-point. Track revision mode was used for all changes in the revised manuscript.
Dr. Sujiang Zhang on behalf of all co-authors
Response to Reviewer 2 Comments |
||
1. Summary |
|
|
We really appreciate your helpful comments and valuable suggestions. Thanks for giving us this precious opportunity to improve the quality of this manuscript. Please find the detailed responses below and the corresponding revisions in track changes in the re-submitted files. Dr Sujiang Zhang on behalf of all co-authors |
||
2. Questions for General Evaluation |
Reviewer’s Evaluation |
Response and Revisions |
Does the introduction provide sufficient background and include all relevant references? |
Yes |
Changed and improved. |
Are all the cited references relevant to the research? |
Yes |
Changed and improved. |
Is the research design appropriate? |
Yes |
Changed and improved. |
Are the methods adequately described? |
Yes |
Changed and improved. |
Are the results clearly presented? |
Yes |
Changed and improved. |
Are the conclusions supported by the results? |
Yes |
Changed and improved. |
3. Point-by-point response to Comments and Suggestions for Authors Comments 1: Line 16-17 and 23-25: use, instead of ; Response 1: we have been revised following the suggestion and replaced” This study aimed to evaluate the fermentation quality, chemical composition, aerobic stability and in vitro digestibility of silage mixtures with different sweet sorghum (SS) and aerial parts of licorice (LC) ratios.” with “The degradation rate of DE, ME, Neg, DM, CP, ADF, NDF and ADL of 50%SS in rumen of sheep were significantly higher than others (P < 0.05).”(line 17-18 and line 31-32) Comments 2: Line 33: avoid repeating area Response 2: we have been revised following the suggestion and modified “ Southern Xinjiang is the largest saline soil area and desertification area in China, which has relatively little natural rainfall, poor soil. ” (“line 33” into “line 42-43”). Comments 3: Line 45: omit ) Response 3: we have been revised following the suggestion and replaced “ (National Research Council, NRC), 2007).” with “[8]”. (line 57) Comments 4: Line 92-93: the sentence is not clear, the temperature was 20 and 25 °C? use at room temperature of 20° C and 25 °C and omit respectively Response 4: Because of the large temperature difference outside, it causes the indoor temperature to fluctuate, so it has a range.(line 121) Comments 5: Line 161: filtered how? the rumen fluid solution 1:2 was used as incubation media? Please describe briefly. Response 5: â‘ line 203-204: we have been revised following the suggestion and replaced “Rumen fluid was immediately filtered, moved to the laboratory, and stored at 39°C in a water 161 bath.” with “The rumen fluid was immediately filtered using four layers of medical gauze, transferred to the laboratory and stored in a water bath at 39°C.” Response 5: â‘¡Lin 207: we have been have been revised following the suggestion and replaced’’1:2”with”1:2 (Rumen fluid: Artificial rumen fluid) ” Artificial rumen fluid preparation The artificial buffer solution was prepared following the method of Menke and Steingass (1988) . Distilled water (400 mL) was added to the HH-S4 type digital constant temperature water bath, followed by the addition of trace element solution (Solution A) (0.1 mL), buffer solution (Solution B) (200 mL), constant element solution (Solution C) (200 mL), indicator solution (1 mL), and reducing agent solution (40 mL). The mixture was preheated at 39°C and saturated with carbon dioxide until it became colorless. The specific preparation method for the artificial buffer solution is as follows: Solution A: Weighed 13.2 g of CaCl2·2H2O, 10.0 g of MnCl2·4H2O, 1.0 g of CoCl2·6H2O, and 8.0 g of FeCl3·6H2O. Dissolved them in distilled water to make a final volume of 100 mL. Solution B: Weighed 4.0 g of NH4HCO3 and 35.0 g of NaHCO3. Dissolved them in distilled water to make a final volume of 1000 mL. Solution C: Weighed 5.7 g of Na2HPO4, 6.2 g of KH2PO4, and 0.6 g of MgSO4·7H2O. Dissolved them in distilled water to make a final volume of 1000 mL. Indicator Solution: Prepared a 0.1% solution of thymolphthalein by weight and volume. Reducing agent solution: Weighed 4 mL of NaOH solution with a concentration of 1 mol/L and 0.625 g of Na2S·9H2O. Dissolved them in distilled water to make a final volume of 100 mL. Two hours before morning feeding, rumen fluid was collected, mixed well, filtered, and collected in a preheated 39°C insulated container. It was quickly brought back to the laboratory and mixed with the prepared buffer solution in a 1:2 ratio to produce artificial rumen fluid. Comments 6: Line 235: not only 100 % but also 75% is significantly different Response 6: line 290: we have been revised following the suggestion and replaced ”100% SS had significantly higher” with “100% SS and 75%SS had significantly higher” Comments 7: Line 243: why you do not report the statistical differences for LAB Response 7: line 299-300: At the end of the sentence, we have been revised following the suggestion and added the words " but Lactic acid bacteria showed a significant upward trend (P < 0.05)" Comments 8: A general comment on the tabs 5 and 6. They would be more legible if there was a line between an item and the following Response 8: In Tables 5 and 6 the appropriate lines have been added. Comments 10: Line 398: the 50% reaches the plateau before the 75%. The figure 2 would be more comprehensible is the x-axes had the real incubation times Response 9: line 398: Put the horizontal coordinates have been added and the graphic has been recreated Comments 10: Line 368: do not use capital letters for with Response 10: Line 442: we have been revised following the suggestion and replaced ”With” with “with”. Comments 11: Line 526-528: try to rephrase… this… this… is not clear Response 11: line 623-625: we have been revised following the suggestion and replaced “The results of this study are similar to those of this study, with the increase of sweet sorghum, the dry matter degradation rate gradually increases.” with “The results of this study are consistent with previous research indicating that an increase in sweet sorghum leads to a gradual increase in the rate of dry matter decomposition.” Comments 12: Line 617: include pages Response 12: line 1047: we have been revised following the suggestion and replenished (line 1345). 4. Response to Comments on the Quality of English Language Point 1: None. Response 1: The English language of the manuscript has been carefully checked and corrected. 5. Additional clarifications Other modifications can be found in track changes in the re-submitted files. |

Reviewer 3 Report
Comments and Suggestions for Authors
Overall comments,
The study is interesting and useful for further research
However, please changes the following points.
literature misquoted
the goal is not clear
Figure 2. Wouldn't it be better to show how gas production changes in 1, 2, 4, 6, 8, 10, 12, 24, 36, 48, 60, and 72 h as in the methodology?
and perform statistics? There are definitely differences between the 100% and 0% groups, but from what time?
Expand - Conclusions. Provide more details
Author Response
Dear the Agriculture team,
We would like to thank you and the reviewers for taking time to review the present manuscript. The comments and suggestions are greatly appreciated, and the manuscript has been markedly improved after addressing the issues raised. All contents in the manuscript have been checked and revised carefully, and reviewers’ questions and suggestions have been responded point-by-point. Track revision mode was used for all changes in the revised manuscript.
Dr. Sujiang Zhang on behalf of all co-authors
Response to Reviewer 3 Comments |
||
1. Summary |
|
|
We really appreciate your helpful comments and valuable suggestions. Thanks for giving us this precious opportunity to improve the quality of this manuscript. Please find the detailed responses below and the corresponding revisions in track changes in the re-submitted files. Dr Sujiang Zhang on behalf of all co-authors |
||
2. Questions for General Evaluation |
Reviewer’s Evaluation |
Response and Revisions |
Does the introduction provide sufficient background and include all relevant references? |
Yes |
Changed and improved. |
Are all the cited references relevant to the research? |
Yes |
Changed and improved. |
Is the research design appropriate? |
Yes |
Changed and improved. |
Are the methods adequately described? |
Yes |
Changed and improved. |
Are the results clearly presented? |
Can be improved |
Changed and improved. |
Are the conclusions supported by the results? |
Can be improved |
Changed and improved. |
3. Point-by-point response to Comments and Suggestions for Authors |
||
Comments 1: literature misquoted. |
||
Response 1: The literature has been checked and corrected for errors. It has also been renumbered according to the journal's requirements. |
||
Comments 2: the goal is not clear. |
||
Response 2: Some modifications and additions have been made to the abstract and introduction in the manuscript. |
||
Comments 3: Figure 2. Wouldn't it be better to show how gas production changes in 1, 2, 4, 6, 8, 10, 12, 24, 36, 48, 60, and 72 h as in the methodology? |
||
Response 3: L398: The Figure 2 has been revised following the suggestion. |
||
Comments 4: and perform statistics? There are definitely differences between the 100% and 0% groups, but from what time? |
||
Response 4: Line 396-398: In the last sentence of the paragraph (result: Subsection 3.6), add “There was a clear difference between the 100% and 0% groups. However, the plateau of gas production for 100% SS occurred at 60h and for 0% SS at 36h.”. |
||
Comments 5: Expand - Conclusions. Provide more details |
||
Response 5: L685-692: The conclusion has been replaced with “In conclusion, the research on the fermentation quality, aerobic stability, degradation characteristics of nylon bag and in vitro degradation characteristics of mixed silage of the whole of sweet sorghum and licorice with stems and leaves in different proportions, which showed that the mixed silage of the whole of sweet sorghum and licorice with stems and leaves in 50:50 (50%SS) effectively improved fermentation quality and rumen degradation rate. The addition of licorice with stems and leaves could effectively improve the aerobic stability and CP content of their mixed silages.” |
||
4. Response to Comments on the Quality of English Language |
||
Point 1: None. |
||
Response 1: The English language of the manuscript has been carefully checked and corrected. |
||
5. Additional clarifications |
||
Other modifications can be found in track changes in the re-submitted files. |

Round 2
Reviewer 3 Report
Comments and Suggestions for Authors
The authors have improved the work. I have no comments. Accept in present form